# URJC-1: Stable and Efficient Catalyst for O-Arylation Cross-Coupling

**DOI:** 10.3390/nano14131103

**Published:** 2024-06-27

**Authors:** Elena García-Rojas, Pedro Leo, Jesús Tapiador, Carmen Martos, Gisela Orcajo

**Affiliations:** Chemical and Environmental Engineering Group, ESCET, Rey Juan Carlos University, C/Tulipán s/n, 28933 Móstoles, Spain

**Keywords:** MOF catalyst, copper MOFs, O-arylation, aldehyde, phenol, fine chemistry

## Abstract

The design of metal–organic frameworks (MOFs) allows the definition of properties for their final application in small-scale heterogeneous catalysis. Incorporating various catalytic centers within a single structure can produce a synergistic effect, which is particularly intriguing for cross-coupling reactions. The URJC-1 material exhibits catalytic duality: the metal centers act as Lewis acid centers, while the nitrogen atoms of the organic ligand must behave as basic centers. The impact of reaction temperature, catalyst concentration, and basic agent concentration was evaluated. Several copper-based catalysts, including homogeneous and heterogeneous MOF catalysts with and without the presence of nitrogen atoms in the organic ligand, were assessed for their catalytic effect under optimal conditions. Among the catalysts tested, URJC-1 exhibited the highest catalytic activity, achieving complete conversion of 4-nitrobenzaldehyde with only 3% mol copper concentration in one hour. Furthermore, URJC-1 maintained its crystalline structure even after five reaction cycles, demonstrating remarkable stability in the reaction medium. The study also examined the impact of various substituents of the substrate alcohol on the reaction using URJC-1. The results showed that the reaction had high activity when activating substituents were present and for most cyclic alcohols rather than linear ones.

## 1. Introduction

Metal–organic frameworks (MOFs) are multimodal organic–inorganic systems formed by organic ligands and metal centers (metal ions or clusters), forming a crystalline pattern and a space-coherent porous lattice with a great diversity in chemical and physical properties [1]. These materials have been used in a variety of environmental and biomedical applications, such as catalysts, sensors, toxic gas and metal ion absorbers, due to their structural properties such as high surface area, high porosity, thermal stability, and pore and lattice functionality [2]. Among the various applications, catalysis is considered to be a mainstay of diverse processes such as the manufacture of commodities, pharmaceuticals, petrochemicals, agrochemicals, food, polymers and cosmetics, and thus about 8000 journal articles and more than 100 patents have been published on the application of MOFs in catalysis over the last decades [3]. The flexibility of MOF materials, achieved through a wide range of organic and inorganic components, makes them highly attractive for fine chemistry [4,5]. However, the number of organic coupling reactions catalyzed by MOF materials are not so common [6,7]. The catalysis landscape has paid significant attention to the development of C-C and C-heteroatom bonds due to their ability to facilitate the formation of high-value products.

In this context, Ullmann cross-coupling is one of the most widely used methods for the economical and safe production of diaryl ethers [8,9,10,11]. This product is a very valuable structural unit, commonly found in a range of important compounds, including pharmaceuticals and natural products, in particular, in pharmaceutically relevant compounds such as Sorafenib, XK469, Tafenoquine, and AMG900 [12,13,14,15,16]. Therefore, the development of methods for the synthesis of diaryl ethers is currently in great demand [17]. In recent decades, the cross-coupling of aryl halides with phenols has been extensively developed and is now considered the conventional route for the synthesis of Ullmann ethers [18,19,20,21]. Although significant progress has been made in the obtention of diaryl ethers by metal-catalyzed homogeneous cross-coupling reactions, the use of a large amount of catalyst together with the continuous increase in precious metal prices [22,23], long reaction time [24,25] and high temperature [26,27], has limited their applications in large-scale syntheses and industrial processes, since homogeneous catalysis suffers from the problematic separation of catalyst from product for reuse [28,29]. Within homogeneous catalysts, copper catalysts have emerged as viable alternatives to palladium-based systems for cross-coupling reactions. This is because they allow the reaction to be carried out without the need for an additional ligand, a capability that palladium catalysts have, which requires the design of specific ligands [30,31]. Considering all of the above, the heterogenization of the existing homogeneous transition metal catalysts appears to be a logical solution to these problems [32,33]. There has been considerable interest in the development of heterogeneous catalytic systems that can be easily recycled while maintaining the inherent activity of the catalytic center, where copper-based catalysts are the most common in the literature [7]. In addition, the use of MOF materials as heterogeneous catalysts has great potential due to their intrinsic properties, and their use in scientific research is still relatively scarce [34]. Therefore, there is a clear need for further research in this area. Metal–organic frameworks (MOFs) offer remarkable flexibility in the development of catalytic systems that facilitate cross-coupling reactions. The metal nodes within MOFs can present free coordination positions that are capable of promoting this type of reaction. However, the nature of the metal is limited to palladium, copper, or other transition metal ions with catalytic activity in organometallic coordination chemistry. A second strategy for developing MOF-based catalysts for cross-coupling reactions has been to utilize the organic linker as a ligand. This strategy can be considered a form of heterogenization, whereby a soluble complex is immobilized on an insoluble solid through the use of the organic component of the MOF as a ligand. These two potential methodologies, when combined with the design of appropriate organic ligands and suitable transition metals, can also readily facilitate the development of bifunctional solid catalysts [35]. In addition, the use of robust MOFs is essential, as these materials must be stable to resist the majority of aggressive solvents and reagents, as well as harsh reaction conditions. In this context, URJC-1 material is based on the ^1^H-imidazole-4,5-tetrazole (HIT) linker and copper (II) ions and exhibits dual acid–base functionality. The uncoordinated nitrogen atoms of the tetrazole groups of the ligand act as basic Lewis centers, while the accessible unsaturated copper sites possess acidic catalytic properties [36]. The presence of the tetrazole groups in the ligand may play a determining role in the catalytic activity of the material since this MOF presents two catalytic centers, the exposed metal centers and the linker as an organocatalytic site. Moreover, the catalytic activity of URJC-1 has been demonstrated in a variety of reactions, including the acylation of anisole, Knoevenagel condensation, and the cycloaddition of CO_2_ with epoxides [36,37]. In this context, URJC-1 is an appropriate candidate for evaluation in O-arylation cross-coupling. In this work, the catalytic potential of this material has been evaluated in the cross-coupling reaction of aryl halides with phenols to obtain diaryl ethers as a product. The influence of different reaction variables was evaluated to determine the best operation conditions for the synthesis of diaryl ethers. In addition, this material was compared to other copper-based MOF heterogeneous catalysts, both with and without the presence of nitrogen heteroatoms in the organic ligand. The reusability of the material was also assessed over multiple reaction cycles.

## 2. Materials and Methods

### 2.1. Materials

All the starting materials and solvents were purchased from Cymit Química, S.L. (Barcelona, Spain) and were used without any further purification.

### 2.2. Synthesis of Organic Linker

The organic linker of URJC-1, ^1^H-imidazole-4,5-tetrazole (HIT), was prepared following the synthetic procedure reported in the work of Leo et al. [36]. The organic linker for HNUST-1, bis(3,5-dicarboxyphenyl) terephthalamide, was prepared according to the method described by Zheng et al. [38]. The organic linker of JUC-62, 3,3′,5, 5′-azobenzene tetracarboxylic acid was prepared following the procedure of Montes-Andrés et al. [39]. Finally, the organic linker of Cu-MOF-74 was purchased from Cymit Química, S.L. (Barcelona, Spain).

### 2.3. Synthesis of MOF Materials

All MOF materials were synthesized by the solvothermal methods previously published for URJC-1 [36], Cu-MOF-74 [40], HNUST-1 [38], JUC-62 [41] and HKUST-1 was purchased from Sigma Aldrich Company, St. Louis, MO, USA.

### 2.4. Physicochemical Characterization Techniques

^1^H NMR spectra were collected with a Varian Mercury Plus spectrometer at 400 MHz using trimethyl silane as an internal standard. FID (flame ionization detector) files were processed using MestRe-C software version 4.9.9.6. The chemical shifts (δ) for ^1^H spectra, given in ppm, are referenced to the residual proton signal of the deuterated chloroform. X-ray powder diffraction (XRD) patterns were acquired on a PHILIPS XPERT PRO diffractometer using CuKα radiation (1.542 Å). The data were recorded from 5 to 50 (2θ) with a resolution of 0.01°. Simultaneous thermogravimetry and derivative thermogravimetric analyses (TGA) were performed under air atmosphere at a heating rate of 5 °C/min up to 800 °C using a SDT 2860 apparatus. Argon adsorption/desorption isotherms were measured at 87 K using 3Flex Micromeritics equipment, prior to the samples being degassed at 150 °C and high vacuum during 12 h. The total surface area was calculated by using the Brunauer–Emmett–Teller (BET) model. Pore volume was assessed using the Dubinin–Radskevich equations. The pore size distribution was estimated using non-local DFT calculations, assuming a kernel model of split pore, Ar-carbon at 87 K. Metal content in the filtered solution after reaction was measured by ICP–OES analysis collected in a Varian VISTA AX system.

### 2.5. Reaction Procedure

URJC-1 was assessed for its catalytic activity in the O-arylation cross-coupling reaction of 4-nitrobenzaldehyde (4-NB) and phenol (Ph) to form 4-formyldiphenyl ether (4-FDE) (Figure 1). The initial conditions to carry out this reaction were derived from previous work, where N,N-dimethylformamide (DMF) as solvent, 120 °C, 4-NB/Ph molar ratio of 1/2, one equivalent of K_2_CO_3_ and 5 mol% of MOF as catalysts were used [42]. The base and catalyst concentrations were adjusted according to molar ratios of base/Ph and Cu/4-NB, respectively. All the catalytic experiments were carried out in a round bottom flask placed in a silicone bath under N_2_ atmosphere. The influence of the reaction temperature, the catalyst concentration, and base concentration were evaluated according to preliminary conditions found in the literature [42,43]. The required amounts of reactants (Ph and 4-NB) were added to 10 mL of the solvent. The reaction was monitored by extracting aliquots from the reaction medium at different times ranging from 0 to 120 min. All reactants and products were identified and quantified by gas chromatography, using a GC-3900 chromatograph with a flame ionization detector (FID). A DB-5 MS Ultra inert capillary column was used as stationary phase of 30 m × 0.25 mm and film thickness of 0.25 µm. All samples were analyzed by three replications and sulfolane was used as an internal standard.

The progress of the reaction was analyzed based on the conversion of 4-NB, which acted as the limiting reagent. The selectivity to diaryl ether FDE was considered 100% as no other by-products were detected. A similar assumption was observed in the literature for this reaction [42,43].

## 3. Results and Discussion

### 3.1. Characterization of MOF Materials

Figure 1 shows the organic linkers selected to synthetize the different copper-based MOFs. URJC-1, JUC-62, and HNUST-1 organic linkers were synthesized following the organic routes mentioned in the experimental section, while Cu-MOF-74 organic linker and HKUST-1 material were commercially purchased. As previously demonstrated, the distinctive organic composition of these ligands enables them to employ a multitude of coordination mechanisms with copper ions [37]. The planar square shape of HKUST-1, JUC-62, and HNUST-1 is formed by the organic ligands being monodentately coupled to Cu^2+^ ions through the oxygen atoms of the acid groups. The nitrogen atoms of the organic ligand HIT monodentately coordinate the copper ion in URJC-1, resulting in the formation of a square-based pyramidal shape. Finally, Cu-MOF-74 contains two distinct copper ions. One is monodentately coordinated to an acid group oxygen atom, while the other is bidentately chelated to an acid group oxygen atom as well as a hydroxyl group oxygen atom. In addition, it is important to consider not only the coordination mode of copper but also the functional organic groups that belong to the linkers. The hydroxyl groups present in the linker of Cu-MOF-74 can be considered Lewis basic groups, in contrast to HKUST-1, which lacks any functional organic group. Furthermore, another three materials were chosen to evaluate the nature of nitrogen present in the organic ligand, tetrazole, azo, and amide groups in URJC-1, JUC-62, and HNUST-1, respectively. This is the first time they have been evaluated in this reaction. It is noteworthy that all the anchor positions of the organic ligands to the metal center are carboxyl groups, whereas in the HIT ligand, the anchor positions are azole and tetrazole groups.

To ensure their correct synthesis of the organic molecules, free of impurities, they were analyzed by ^1^H NMR (Appendix A). These organic linkers were then used to crystallize the corresponding MOF structure and the presence of the pure phases was verified using powder X-ray diffraction. The XRD patterns of the four synthetized MOF materials were compared with their corresponding simulated XRD patterns from single crystal XRD in the Supporting Information (Appendix A). Finally, HKUST-1 was acquired commercially but was also characterized by XRD to confirm its correct phase (Appendix A). In all cases, the presence of exclusively the diffraction peaks of each phase in the experimental patterns confirmed the purity of the desired MOF in the bulk samples.

Thermogravimetric analyses (TGA) under air atmosphere were performed for the five MOF materials to determine their thermal stability and the appropriate outgassing temperature (Appendix A). According to TGA profiles, 110 °C was the activation temperature for JUC-62, 120 °C for Cu-MOF-74, HKUST-1 and HNUST-1 materials and 150 °C for URJC-1. Argon adsorption/desorption isotherms were performed to evaluate their textural properties (Appendix A), and BET surface area, pore volume, and pore diameter are summarized in Table 1. In all cases, textural proprieties match well with those reported in the literature, confirming the obtaining of each copper crystalline material as well.

### 3.2. Catalytic Study

Coupling reactions require a base to achieve effective results; carbonates or phosphates are often used because of their high solubility and low cost [44]. Some previous experiments have demonstrated that the cross-coupling reaction can only be achieved through the combined action of a base and a catalyst [45], so K_2_CO_3_ was chosen as the base in this work, which role should be to activate the phenol, producing a phenolate ion necessary to boost the reaction. When the reaction was carried out without a catalyst but with the base, a surprisingly high 4-NB conversion was achieved. However, the target 4-formyldiphenyl ether (4-FDE) product of cross-coupling was not detected. In this case, a solid compound was observed, which was characterized by ^1^H NMR (Appendix A). Based on this result, we propose that this solid is the result of the undesired Cannizzaro reaction, such as a 4-nitrobenzoate salt (Scheme S6.2). On the other hand, when the reaction was performed with a catalyst without K_2_CO_3_, only around 10% 4-NB conversion was observed, similar to that obtained in the blank reaction, but no presence of Cannizzaro products was detected, demonstrating the need for deprotonating the phenol first to favor the reaction.

Since URJC-1 exhibits dual properties in catalysis from its both acidic and basic centers in the same material, it was selected to study the principal reaction variables, starting with the best conditions obtained in a previous work. This reaction was carried out at 120 °C using N-dimethylformamide (DMF) as the solvent, with an NB/Ph molar ratio of 1/2, one equivalent of K_2_CO_3_, and 5 mol% of catalysts [42]. The initial rate of the reaction was remarkable, with 85% conversion of 4-NB achieved in just 15 min and reaching 100% conversion in 90 min.

#### 3.2.1. Influence of Catalyst Concentration

The catalyst concentration was the first reaction variable tested for the conversion of 4-nitrobenzaldehyde. URJC-1 loadings of 0, 2, 3, 4, and 5 mol% MOF were tested, as shown in Figure 2. It is noteworthy that this concentration refers to Cu content with respect to the 4-NB substrate. A reduction in catalyst concentration from 4 mol% resulted in slower kinetics at the beginning of the reaction. However, after 60 min, roughly the complete conversion was achieved using any catalyst concentration, except for 2 mol% catalysts, where the 4-NB conversion did not exceed 90% even after 2 h. So, a catalyst loading of 3 mol% was chosen as optimal for further studies. It is worth noting that the copper content required for the complete conversion of 4-NB is significantly lower than conventional Ullman reactions, which require an average of 10 mol% Cu catalyst concentration [46].

URJC-1 has been demonstrated to be a highly robust heterogeneous catalyst in various test reactions, such as Friedel–Crafts acylation or Knoevenagel condensation, exhibiting exceptional thermal and chemical stability [36]. As before, we observed herein the same behavior, confirming that its crystalline structure remains intact after the reaction regarding the X-ray diffraction analysis, shown in Figure 3a. However, it is possible that the copper species present in the metal–organic structure may dissolve in the solution, leading to either homogeneous or partially homogeneous catalytic conditions for cross-coupling [47]. To test these hypotheses, an additional catalytic run was carried out at 120 °C with 3 mol% of URJC-1. The solid catalyst was removed from the mixture by hot filtration after 15 min of reaction (hot filtration test). Figure 3b displays the 4-NB conversion profile after removing the catalyst, which is evidenced by the sudden stoppage of the reaction, which confirms that the heterogeneous URJC-1 catalyst plays an active role in the O-arylation cross-coupling reaction. Moreover, since no further conversion of 4-NB was observed after removing the solid catalyst from the reaction mixture, it indicated there was no contribution from homogeneous catalysis due to plausible copper leaching. Additionally, no dissolved copper species were detected in the reaction mixture according to ICP-OES analyses.

#### 3.2.2. Influence of Base Concentration

The concentration of K_2_CO_3_ base was evaluated in the range of 0.5–2.0, remaining constant the rest of the reaction conditions: DMF as a solvent, 120 °C, 4-NB/Ph = 2, and a catalyst loading of 3 mol%. Figure 4 shows that the highest conversion is obtained when one equivalent of base is used, which seems to be enough for efficient deprotonation of phenol, as was stated previously in other works [42,43]. Increasing to two equivalents did not lead to any significant improvement in catalytic activity. When 0.75 and 0.5 equivalents are used, the deprotonation of phenol does not appear to be complete, thus reducing the 4-NB conversion achieved to 75%.

#### 3.2.3. Influence of Temperature

The influence of temperature in the cross-coupling reaction was another parameter studied, and it was evaluated at 140, 120, 100 and 80 °C. These experiments were performed by fixing the rest of the variables at their optimum value (120 °C, 4-NB/Ph = 2, 1 equivalent of K_2_CO_3_, and a catalyst loading of 3 mol%). Figure 5 shows that a decrease in reaction temperature leads to a reduction in 4-NB conversion. On the other hand, an increase up to 140 °C enhances the initial reaction rate, but after one hour the conversion reaches the same value as that obtained at 120 °C (97%), so the last is good enough to reach almost complete conversion at short times, so it was fixed for further analysis. The stability of the material does not seem to be affected by the increase in reaction temperature, as confirmed by XRD and ICP analysis.

#### 3.2.4. Comparison with Other Catalysts

The catalytic behavior of URJC-1 was compared with other copper-based catalysts, including homogeneous and heterogeneous copper ones. The experiments were carried out under the optimum conditions determined in this study: 120 °C, DMF as the solvent, 1 equivalent of K_2_CO_3_, 4-NB/Ph molar ratio of 1/2, and a catalyst concentration of 3 mol% catalyst. Firstly, the catalytic behavior of URJC-1 was compared to the homogeneous Cu(NO_3_)_2_, to the heterogeneous CuO catalyst. Figure 6 shows the conversion evolution of these three catalysts. Both, the salt and the oxide display a significant conversion, but it is evident that the MOF material leads to a better catalytic activity, both in terms of conversion values and initial reaction rates, so the Cu environment in the MOF structure seems to improve its performance.

The copper-based MOF materials HKUST-1, Cu-MOF-74, HNUST-1, and JUC-62 exhibit an octahedral coordination geometry, with bond distances varying depending on the solvent to which they are coordinated. This causes them to exhibit different structural and chemical properties. In addition, these materials were evaluated in terms of their catalytic performance in comparison with the URJC-1 material (Figure 7). Once more, the highest conversion was obtained when URJC-1 was used as a catalyst. This is particularly interesting since its textural properties are significantly worse than those for the other materials, which have at least twofold specific surface and pore volume, except for HNUST-1 with poorer textural properties. As explained in previous work [36], there is a synergistic catalytic effect between the unsaturated Cu Lewis acid sites and the tetrazole-based ligand in URJC-1 that could contribute to its better performance. In fact, other works have found an increase in cross-coupling efficiency when N-containing ligands were used since they can participate as basic centers in the reaction [48,49,50]. However, the reaction results with other MOFs containing nitrogen atoms in their structure, such as HNUST-1 and JUC-62, with amide and azo groups, respectively, were lower than URJC-1, so not all N-containing ligands led to an improvement in the catalytic performance, as discussed in reference [37]. In addition, these last materials were not stable in the reaction medium, as can be seen in Appendix A. Dissolved copper species were detected in the reaction mixture through ICP-OES analysis, leading probably to homogeneous catalysis.

Table 2 presents a comparison of the works reported in the literature on the O-arylation cross-coupling reaction using copper-based catalysts and MOFs. It is observed that the use of stronger bases for the linkers is necessary to achieve the desired product and that only a few works employed MOFs as catalysts. In comparison, this work employed a smaller amount of catalyst and only 60 min is necessary to reach a 100% conversion.

#### 3.2.5. Recyclability of URJC-1

An important environmental advantage of using MOF materials in this type of reaction is their recovery and subsequent recyclability for multiple reaction cycles. It is also known for the limited stability of MOFs in reaction media. For those reasons, the reusability of URJC-1 in this cross-coupling was evaluated in the reaction conditions used in the previous sections (120 °C, DMF as solvent, 1 equivalent of K_2_CO_3_, 4-NB/Ph molar ratio of 1/2 and 3 mol% catalyst).

URJC-1 presents no significant loss of activity over the five cycles of 60 min of reaction, maintaining a 4-NB conversion above 90% (Figure 8a). To check the stability of the MOF structure after those cycles, XRD patterns were collected for the catalyst, as shown in Figure 8b. It can observe the presence of the URJC-1 crystalline phase almost intact after the reaction, maintaining its principal diffraction peaks. ICP analysis was performed on the reaction media, indicating that there was no leaching of copper. So, the progressive slight decrease in 4-NB conversion after cycles could be related to a slight loss of catalyst mass during the filtration process in each cycle instead of catalyst damage itself. Therefore, URJC-1 has been demonstrated to be a robust and efficient acid–base copper catalyst for O-arylation cross-coupling reactions.

#### 3.2.6. Catalytic Activity of Different Substrates

We extended our catalytic study of URJC-1 to different substrates. Figure 9 shows the 4-NB conversion when 4-chlorophenol, 3-chlorophenol, 3-nitrophenol, 2-nitrophenol, guaiacol, p-cresol, cyanophenol, 1-fluoro-4-nitrophenol and 4-nitrophenone were used. The reactions were carried out at 120 °C, DMF as a solvent, 1 equivalent of K_2_CO_3_, 4-NB/Ph molar ratio of 1/2 and 3 mol% catalyst loading. Several substituents in phenol molecules with different activating and deactivating organic groups have been used in the O-arylation cross-coupling reaction. For methoxy and chloro groups in phenol, higher 4-NB conversions were reached, as can be seen in 1b, 1c and 1d reactions, possibly since they induce a conjugative effect +K, which increases the electron density to the hydroxy group of phenol, enhancing its nucleophilicity. On the other hand, the nitro and cyano groups reduce the electron density of the hydroxy group due to a conjugative effect -K, which causes a decrease in the 4-NB conversion, as shown in the reactions 1e, 1f, and 1g, being the *meta-* position is less deactivate than *orto*- and *para*-position. The size of the molecule is another important factor, for example, in reaction 1h the conversion was slightly lower than in reaction 1a (no substituent), this behavior is related to the steric hindrance generated inside the cavities of the MOF material. In addition, a smaller butanol was also tested, which showed the lowest 4-NB conversion (14%) due to the absence of an aromatic ring in the substrate reducing the kinetic of the reaction. Finally, two different nitro derivatives were used in the reaction, and a similar result was obtained when 1-fluoro-4-nitrobenzalhyde was used in reaction 1i, however, the phenone group reduces the activity due to the -K effect in reaction 1j.

#### 3.2.7. Proposed Mechanism for O-Arylation Cross-Coupling

The reaction mechanism has been proposed by means of different previous works, as shown Figure 10 [55,56]. In the first step, the nitro group of 4-nitrobenzaldehyde interacts with the unsaturated Cu center by one oxygen atom of this group. This interaction produces the polarization of the N-C bond resulting in the elimination of the NO_2_^−^ group, which forms the KNO_2_ salt through the K_2_CO_3_ base (Step II). This elimination produces the coordination of the metal center to the aromatic carbon atom of the 4-nitrobenzaldehyde in Step III. Finally, in the fourth step, the formation of phenolate takes place by means of the potassium carbonate. This new species interacts with the copper center producing the nucleophilic attack of the phenolate group to the aromatic carbon atom of benzaldehyde giving rise to the C-O cross-coupling obtaining the product of the reaction, which is released to the reaction medium generating the catalytic site once again.

## 4. Conclusions

The catalytic behavior of the MOF URJC-1 material has been studied in the O-arylation cross-coupling reaction between 4-nitrobenzaldehyde and phenol. This material possesses metallic copper centers bonded to five nitrogen atoms belonging to the tetrazole and imidazole rings of the ligand, which implies the possibility of having two different types of catalytic centers. In this context, the principal reaction conditions, such as MOF concentration, temperature, and base amount, were optimized. Cross-coupling did not occur in the absence of a catalyst. However, with a catalyst concentration of 3 mol%, the NB was fully converted, and the desired product achieved 100% selectivity within an hour at 120 °C, excluding the leaching of copper species into the reaction medium. Thus, the URJC-1 material showed high reaction stability, maintaining the same crystalline phase. Additionally, other copper-based catalysts such as Cu(NO_3_)_2_, CuO, HKUST-1, Cu-MOF-74, HNUST-1 and JUC-62, were evaluated under these conditions. URJC-1 showed the best catalytic behavior in the reaction studied, always taking into account the conversion and the stability of the structure. Considering the results obtained, it was necessary to evaluate its role as a heterogeneous catalyst during several consecutive reaction cycles, with favorable results. Therefore, the physico-chemical properties of this MOF structure enable a reduction in the required amount of catalyst compared to other materials previously reported in the literature. Therefore, URJC-1 has been demonstrated to be a robust and efficient acid–base copper catalyst for O-arylation cross-coupling reaction, and it should be explored in other similar catalyzed reactions of industrial interest.

## Data Availability

Data is contained within the article or Appendix A.

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
