# Peer review of "URJC-1: Stable and Efficient Catalyst for O-Arylation Cross-Coupling"

_nanomaterials, 2024, doi:10.3390/nano14131103_

Round 1
Reviewer 1 Report
Comments and Suggestions for Authors
In this articles, the catalytic duality of URJC-1 material was demonstrated. The metal centers act as Lewis acid centers and the nitrogen atoms of the organic ligand must behave as basic centers. The effects of reaction temperature, catalyst concentration, and alkaline reagent concentration were evaluated. Several copper based catalysts, including homogeneous and heterogeneous MOF catalysts with and without nitrogen atoms in organic ligands, were evaluated for their catalytic performance under optimal conditions. I suggest accepting after minor modifications. This study is crucial for the application of metal organic frameworks (MOFs) in small-scale heterogeneous catalysis. The paper has the following problems to be improved:.
1. Language polishing is required for this manuscript.
2. The lack of structural testing of the material, such as SEM, TEM, etc. , suggested supplement.
3. The reason why this material has good catalytic performance is lack of analysis, why it has good catalytic degradation effect, is because it will produce active oxygen vacancy? Hydroxyl radicals?
4. Need to supplement the degradation flowchart..
Comments on the Quality of English Language
In this articles, the catalytic duality of URJC-1 material was demonstrated. The metal centers act as Lewis acid centers and the nitrogen atoms of the organic ligand must behave as basic centers. The effects of reaction temperature, catalyst concentration, and alkaline reagent concentration were evaluated. Several copper based catalysts, including homogeneous and heterogeneous MOF catalysts with and without nitrogen atoms in organic ligands, were evaluated for their catalytic performance under optimal conditions. I suggest accepting after minor modifications. This study is crucial for the application of metal organic frameworks (MOFs) in small-scale heterogeneous catalysis. The paper has the following problems to be improved:.
1. Language polishing is required for this manuscript.
2. The lack of structural testing of the material, such as SEM, TEM, etc. , suggested supplement.
3. The reason why this material has good catalytic performance is lack of analysis, why it has good catalytic degradation effect, is because it will produce active oxygen vacancy? Hydroxyl radicals?
4. Need to supplement the degradation flowchart..
5. During the fifth cycle, the height of the first peak of XRD decreased obviously, and the reason should be analyzed.
Author Response
We appreciate the effort made by the editor and reviewers in our manuscript. We have paid attention to every comment, and accordingly we have modified the manuscript, that is now improved. Our answers to their comments are included below.
Please see the attachment.

Reviewer 2 Report
Comments and Suggestions for Authors
Elena García-Rojas and coworkers reported ‘URJC-1: Stable and Efficient Catalyst for O-Arylation Cross- 2 Coupling’. The experimental is carefully conducted, and the results have been presented correctly, and the contents fall well into the scope of the journal. However, some points need to be addressed/answered before further considered:
1. Please describe the mechanism of catalytic conversion in detail and use theoretical simulations to further support your results.
2. Figure 3a: it is hard believe that the XRD patterns of URJC-1 before and after reaction are almost the same.
3. Please cite the following two papers: “Micropor. Mesopor. Mater., 2022, 112098; “J. Mater. Chem. A, 2020, 8, 11933”.
Author Response

(The authors gave the same response as above.)

Reviewer 3 Report
Comments and Suggestions for Authors
The authors report the use of the URJC-1 MOF as catalyst for the O-arylation of phenols. The stability and the reusability of the Cu-based MOF were demonstrated. The work is of interest but the scope of the reaction seems rather limited. The following comments should be considered by the authors :
- results could be better discussed in the context of literature.
- a mechanism should be proposed for the Cu-catalyzed O-arylation reaction.
- the authors must compare the performance of the URJC-1 catalyst for the preparation of formyl ethers to other catalyst described in the literature (not only to Cu-based catalysts as described in paragraph 3.2.4) and highlight the advances made.
- can be coupling be conducted using other aromatic nitro derivatives ? Justify the reactants used (4-NB and phenol) to conduct the coupling.
- Figure 7 : are the differences in performance observed for the various MOFs significative ? How many times were the couplings performed ?
- line 300 : correct "ciano" into "cyano".
- Results described in Figure 9 must be better discussed. For example, the presence of the electron withdrawing CN group (entry 2g) does not significantly reduce the efficiency of the coupling.
Comments on the Quality of English Language
Only minor changes are required.
Author Response

(The authors gave the same response as above.)

Reviewer 4 Report
Comments and Suggestions for Authors
The paper "URJC-1: Stable and Efficient Catalyst for O-Arylation Cross- Coupling" reports on the evaluation of the catalytic efficiency of URJC-1 MOF in Ullmann coupling reaction between phenol and different substituted aromatic aldehydes. The idea of the manuscript is well designed and in general the results support the novelty of this study. However, there are some issues regarding the presentation of the data.
-Abstract-the first two sentences must be rewritten or deleted,.
-Introduction: The catalytic activity of the URJC-1 is known in different reactions, these data must be added. The motivation of choosing these MOFs must be better highlighted. The authors also evaluated the catalytic effect of other MOFs, but the introduction is focused only on URJC-1.
Section 2.4. -should include all characterization methods used by authors, but some of these are found in section 2.5 at lines 117-120 and 145-146. This section must be rewritten.
Section 3.1. - The fig. 1 should refer only to the linkers used in the synthesis of the MOFs presented in each case, the differences must be shown. However, the structure of the ligands is quite different from URJC-1, but the catalytic efficiency is comparable. In this case, the acido-basic structure of the MOFs is consistent?
Section 3.2.1. It is visible that the concentration of the catalyst does not influence the conversion rate in a visible way.
Even the conversion product was emphasized by GC, the isolation and the full characterization, at least by NMR or MS must be proved.
As a general observation: too many figures are in Supplementary material. At least the structure of the MOFs as the authors have obtained must be presented, along with the structural differences as XRD analysis revealed. All MOFs prepard by the authors possess the same numbe of copper ions coordinated in a similar geometry? Some discussions must be added.
I recommend Major revision for this manuscript before the acceptance.
Author Response

(The authors gave the same response as above.)

Round 2
Reviewer 3 Report
Comments and Suggestions for Authors
All corrections were made. The manuscript can be accepted by Nanomaterials.
Reviewer 4 Report
Comments and Suggestions for Authors
The authors revised the manuscript according to the suggestions and can be accepted in the present form.